# Factors associated with viral load re-suppression after enhanced adherence counseling among people living with HIV with an initial high viral load result in selected Nigerian states

Gbenga Benjamin Obasa[1]*, Mukhtar Ijaiya[1], Ejike Okwor[1], Babafemi Dare[1], Franklin Emerenini[2]*, Remi Oladigbolu[1], Prince Anyanwu[2], Adewale Akinjeji[2], Kate Brickson[3], Jennifer Zech[4], Yemisi Ogundare[1], Emmanuel Atuma[1], Molly Strachan[3], Ruby Fayorsey[4], Kelly Curran[3]

1 Jhpiego, Ankuru, Nigeria, 2 ICAP Global Health, Abuja, Nigeria, 3 Jhpiego, Baltimore, Maryland, United States of America, 4 ICAP Global Health, New York, New York, United States of America

☯ These authors contributed equally to this work.
‡ These authors also contributed equally to this work
* gbo2105@cumc.columbia.edu (GBO); fe2204@cumc.columbia.edu (FE)

## Abstract

The WHO recommends monitoring viral load (VL) to gauge ART efficacy among People Living with HIV (PLHIV). Low suppression rates persist in low- and middle-income countries due to poor adherence. Enhanced Adherence Counseling (EAC) aims to improve adherence and treatment outcomes. This study, part of the Reaching Impact Saturation and Epidemic Control (RISE) project in Nigeria, analyzes factors affecting viral re-suppression post-EAC. It aims to inform clinical decisions and improve PLHIV health outcomes in the country. This was a retrospective analysis of a de-identified client-level dataset of unsuppressed VL clients who were current on treatment at the end of June 2022 and subsequently enrolled in the EAC program. A log-binomial regression model was used to report crude and adjusted risk ratio with 95% Confidence Intervals (95% CI) and a p-value of 0.05 to determine the association between clinical characteristics and suppression of VL post-EAC in the RISE program (July 2021 to June 2022). A total of 1607 clients with initial high VL who completed EAC were included in this analysis out of which 1454 (91%) were virally suppressed. The median time to completion of EAC was 8 weeks and the median time for post EAC VL test was 8 weeks. Following EAC, PLHIV in the 10–19 years age band were 10% more likely to be re-suppressed (ARR: 1.10; 95% CI 1.01 to 1.19). In addition, there was a 50% reduced likelihood of viral re-suppression among PLHIV on second-line regimens compared to PLHIV on first-line regimens (ARR: 0.50; 95% CI 0.41 to 0.62). Findings show that Age and ART regimen were significant predictors of VLS. More targeted outreach of EAC amongst second-line regimens and ages 10 and above is necessary to ensure better VLS within these groups.

**Data Availability Statement:** All data underlying the results presented in the study are within the paper and its Supporting Information files.

**Funding:** This analysis required no funding. GBO, MI, EO, BD, KB, YO, EA, MS, and KC are Jhpiego employees, and FE,RO, PA, AA, KA and RF are, employees of ICAP at Columbia University; both groups support the Jhpiego-led Reaching Impact, Saturation, and Epidemic (RISE) Control consortium (Grant Number: 200AA19CA00003) funded by the U.S. President's Emergency Plan for AIDS Relief (PEPFAR) through the United States Agency for International Development (USAID). USAID/PEPFAR had no role in the study design, data collection and analysis, and interpretation of the data. The views expressed in this article are those of the authors and not USAID/PEPFAR.

**Competing interests:** The authors have declared that no competing interests exist.

# Background

The World Health Organization (WHO) recommends monitoring Viral load (VL) outcomes as the gold standard to measure the effectiveness of antiretroviral therapy (ART) for people living with HIV (PLHIV) [1,2]. In 2021, WHO revised its HIV treatment monitoring algorithm to assist PLHIV in reaching viral suppression and maintaining an undetectable VL with three categories of VL level: unsuppressed ($>$ = 1000 copies/mL), suppressed (detected but $<$1000 copies/mL) and undetectable (VL not detected by the test used) [3]. VL is considered unsuppressed when $>$ = 1000 copies/ml for PLHIV undergoing treatment for more than six months. WHO recommends VL testing for PLHIV on treatment for more than six months and then routinely every 12 months [1,4]. However, there are reports of low VL suppression rates among PLHIV living in low- and middle-income countries [5,6]. Poor adherence to HIV treatment is the most common reason for viral non-suppression and can potentially increase the risk of drug resistance, HIV treatment failure, and onward HIV transmission [7,8].

The WHO recommends Enhanced Adherence Counseling (EAC) for PLHIV with unsuppressed VL ($>$ = 1000 copies/ml) after six months of ART [2]. The EAC sessions lasting between 3–6 months are designed to improve adherence, identify possible barriers to treatment, and resolve treatment difficulties [5]. Studies have shown that informing PLHIV about the risks of an unsuppressed VL result can improve treatment adherence [9]. This will reduce the chances of developing resistance mutations, treatment failure, or disease progression to advanced HIV disease [10]. The combination of EAC with routine VL has improved outcomes among people on ART [10–12]. A first-line treatment failure is established when VL suppression is not attained after EAC sessions. Subsequently, switching to second-line treatment is considered as different ART classes are associated with varying adherence thresholds needed to achieve viral suppression and avoid resistance mutations [13].

Nigeria has the second largest HIV epidemic in the world, with a total of 1.9 million (1.4%) persons (15–49 living with HIV as reported by the Nigeria HIV/AIDS Indicator and Impact Survey (NAIIS) [14–16]. In 2020, there were around 1.8 million HIV/AIDs PLHIV, and about 67% of all people living with HIV knew their status, but only 62% were on HIV treatment, of which 44% were estimated to be virally suppressed [16]. This VL suppression is sub-optimal compared to the UNAIDS third 95 target [15,16]. Studies have suggested several factors associated with VL non-suppression, such as poor medication adherence, suboptimal drug selection, and drug resistance [3,5,9]. Therefore, the need to assess the effectiveness of EAC in routine program implementation and determine factors associated with poor VL suppression is essential.

This analysis was conducted as part of the Reaching Impact, Saturation, and Epidemic Control (RISE) project, a five-year global project funded by the US President's Emergency Plan for AIDS Relief (PEPFAR) through the US Agency for International Development (USAID). We analyzed factors associated with viral re-suppression for PLHIV who enrolled and completed all EAC sessions between July 2021 and June 2022. We analyzed the demographic and clinical characteristics of PLHIV with unsuppressed VL ($>$ = 1000 copies/ml) who had completed EAC. This analysis will inform ongoing programmatic and clinical decision-making and improve health outcomes for PLHIV in Nigeria.

# Methodology

## Study design

This study was a retrospective cohort study, using routinely collected service delivery data of PLHIV on ART in 132 RISE-supported health facilities from July 2021 to June 2022.

## Program setting and description

In Nigeria, the provision of HIV care is well-established across a network of health facilities, spanning from primary health centers to tertiary care institutions. These facilities offer comprehensive services for HIV prevention, testing, treatment, and care, adhering to guidelines set by the National Agency for the Control of AIDS (NACA) and the Federal Ministry of Health (FMOH). Antiretroviral therapy (ART) programs, in alignment with WHO recommendations, serve as the cornerstone of HIV care, ensuring free access to ART medications and regular clinical assessments. Integrated within various healthcare services, including maternal and child health clinics, HIV care aims to provide patient-centered support, promote optimal health outcomes, and enhance the quality of life for individuals living with HIV/AIDS.

According to the NAIIS data Akwa Ibom and Rivers States exhibit the highest HIV prevalence rates in Nigeria [15,17]. Akwa Ibom and Cross River states are located in the country's South-South region and have a prevalence rate of 4.8% and 1.8%, respectively [15]. In 2019, the RISE project was initiated to ensure treatment saturation and epidemic control within these regions alongside other States in the northeast, Adamawa and Taraba having HIV prevalence rates of 2.6 and 1.1%, and Niger in the north-central zone having a prevalence of 0.9% [15].

The RISE project spans 132 facilities across 56 local government areas in five Nigerian states: Adamawa, Akwa Ibom, Cross River, Niger, and Taraba. Initially launched in four states —Adamawa, Akwa Ibom, Cross River, and Niger—the project integrated seamlessly with existing healthcare infrastructure by leveraging facilities previously supported by other implementing partners. In October 2021, the program expanded to include Taraba State. The Enhanced Adherence Counseling (EAC) Program, introduced concurrently, achieved an initial viral load suppression rate of 79% by June 2021. To optimize EAC utilization, healthcare workers and case managers underwent comprehensive training on EAC implementation.

RISE provides EAC for virally unsuppressed PLHIV enrolled in ART services at supported facilities in alignment with WHO guidelines. EAC is also offered to PLHIV who receive drugs outside the facility at the community level. Children and adolescents (<15 years) with an unsuppressed VL are also enrolled in the EAC program, with their caregivers attending the EAC sessions to assist with treatment adherence and VL suppression. According to WHO and Nigerian national guidelines, PLHIV with high VL results requires a repeat test after at least three EAC sessions. People living with HIV with VL results $> = 1000$ copies/ml are enrolled into the EAC and closely monitored frequently with a minimum of 10 contacts monthly for at least three months, which can be extended to a maximum of 6 months to assess adherence level, after which a VL test is carried out post-EAC. The EAC session is designed to help PLHIV identify the barriers they face in taking their treatment and formulate an adherence plan. Counselors help PLHIV identify their goals and provide emotional support to help them reach them. After completion of the EAC sessions, if the VL result is <1000 copies/ml post-EAC, the PLHIV remains on the same ART regimen. If the VL result is $> = 1000$ copies/ml after completion of the EAC sessions, the PLHIV is switched to a second-line ART regimen.

## Program data

De-identified client-level data from the Lafiya Management Information System (LAMIS), a web-based electronic medical system, was used to identify PLHIV on ART with VL $> = 1000$ copies per ml. Client information included in the data set were sex, date of birth, ART start date, days of ART refill, last drug pickup date, ART regimen type, date of VL testing, WHO clinical staging at last visit, tuberculosis (TB) status at last visit, date of commencement of EAC, EAC enrollment, date of last EAC Session and repeat VL result after completion of EAC.

Data cleaning was carried out, for instance, by converting the birth date to age and then breaking it down into age-group categories. The variable "duration on ART" was derived from the difference between the start date of ART and the date of the last drug pickup. Our outcome variable (VL result post-EAC) was then encoded as a binary classification, which indicated whether the patient was unsuppressed (VL $> = 1000$ copies/ml) or resuppressed (VL detected $< 1000$ copies/ml).

## Study population

The study population for the Enhanced Adherence Counseling (EAC) intervention encompassed individuals of various demographic groups, including children, adolescents, women, and the elderly, receiving care in 132 health facilities located across Adamawa, Akwa Ibom, Cross River, Taraba, and Niger. Specifically, the population consisted of PLHIV who were identified as having unsuppressed VL and who had undergone the EAC intervention as part of their HIV care continuum between January 2020 and December 2021 in these facilities. The PLHIV in this study either received ARV in the facility or the community within each State.

**Inclusion criteria.** All PLHIV enrolled in HIV care through the RISE project who have received ART for at least six months have VL results $> = 1000$ copies/ml and who started and completed EAC sessions (at least three sessions) between July 2021 and June 2022.

**Exclusion criteria.** All PLHIV with no VL result, VL sample collection date, or PLHIV whose VL result was greater than one year were excluded. Also, PLHIV with data outside the selected period with no record of EAC session and documented post-EAC VL results were excluded.

## Data analysis

Data were analyzed using SAS On-Demand version 9.04. The primary outcome of the analysis was VL suppression ($<1000$ copies/ml) after the completed EAC session. The analysis variables were sex, age, enrollment settings, clinical staging, TB co-infection, ARV regimen line, and duration on ART. Descriptive analyses were carried out to show demographic and clinical characteristics of PLHIV with unsuppressed VL ($> = 1000$ copies/ml), suppressed but detectable ($> = 50$ to $<1000$ copies/ml), and undetectable VL ($<50$ copies/ml). Pearson's Chi-Square was used to compare demographics, clinical characteristics, and VL re-suppression, using a significant level of $p<0.05$. Furthermore, the Log binomial regression model was used to report crude and adjusted risk ratios with 95% Confidence Intervals (95% CI) to determine the association between clinical characteristics and VL re-suppression ($<1000$ copies/ml) on repeat testing, post-EAC.

**Ethical statement.** This retrospective analysis adhered strictly to ethical guidelines, ensuring participant privacy and confidentiality. Medical records were accessed for research purposes on October 26th, 2022, with rigorous de-identification measures implemented to safeguard individual identities. Throughout the study, the research team did not have access to personally identifiable information, and all data were anonymized before analysis. This de-identification process involved removing any personal identifiers and managing indirect identifiers that could potentially lead to participant identification. Following data analysis, archived samples and records were securely stored. Given the utilization of secondary data analysis, consent was waived by the ethics committee, considering the retrospective and anonymized nature of the data for both adult and minor clients.

## Ethical approval

The study received ethical approval from Johns Hopkins School of Public Health Institutional Review Board IRB No.00008634 "Secondary Data Analysis of HIV Prevention, Care and

Treatment Service Delivery Programs in Africa, which explicitly reviewed and approved the de-identification procedures to safeguard participant confidentiality.

## Results

A total of 127,198 PLHIV were on ART at RISE-supported Health Facilities. Among this, 5,578 PLHIV with unsuppressed Viral Load (VL) were identified and enrolled in the EAC program as of July 2021. Of these participants, 1,693 individuals (30%) completed the first EAC session, followed by 728 individuals (13%) who completed the second session. A total amount of 3,157 PLHIV (57%) had completed all three EAC sessions. Subsequently, post-EAC VL testing was conducted for those who completed the third session, of which only 1,607 PLHIV (51%) had documented VL results following the EAC. The distribution of PLHIV with VL results following the EAC across the states was as follows: Adamawa 272 (16.9%), Akwa Ibom 321 (19.9%), Cross River 304 (18.9%), Niger 617 (38.4%), and Taraba 92 (5.73%) respectively.

Among these 1,607 participants with documented VL results after EAC, 1,054 (66%) were female. The majority, 1,410 (88%), were enrolled in facility settings, and 1,506 (88%) were on a first-line ART regimen at their last visit. Most individuals 1537 (96%) were classified as WHO clinical-stage 1 at baseline, and 1,557 (97%) were free from TB co-infection. PLHIV aged 40 and above accounted for 589 (37%), while children < 10 years accounted for 7% (114) of PLHIV included in this analysis. Approximately 22% (361) of the PLHIV included in this analysis had been on ART for two years or less. The median time taken to complete all recommended EAC sessions was 12 weeks (Table 1). Over 90% of participants completed all recommended EAC sessions within three months. Following the completion of EAC sessions, the median time for collection of a repeat VL test was 8 weeks. About 32% of the participants had a post-EAC VL test done after 8 weeks

After the completion of EAC, a total of 1,454 individuals (90%) achieved a suppressed VL (<1000 copies/ml) (Table 2). Among them, 336 (23%) had VL > = 50 - < 1000 copies/ml, and 1,118 (76%) had an undetectable VL. Females comprised 66% of those who achieved VL re-suppression after EAC. The age group of 40 years and above had the highest proportion of individuals with re-suppressed VL, totaling 542 (37%), compared to other age groups. In facility settings, 87% of PLHIV achieved VL re-suppression, totaling 1,267 individuals, while in community settings, 13% achieved re-suppression, comprising 187 individuals. The majority of individuals with WHO clinical-stage 1 (91%) attained VL re-suppression after EAC. Additionally, 91% of PLHIV without TB co-infection achieved suppressed VL after EAC ($p<0.001$). Furthermore, 1,408 (93%) PLHIV on the first-line ART regimen achieved VL re-suppression, compared to PLHIV on second-line ART 46 (45%) ($p<0.001$). PLHIV on treatment for more than 3–5 years had a higher proportion of VL re-suppression than those on other durations of antiretroviral therapy ($p<0.031$). Table 2 further details PLHIV characteristics and VL re-suppression in the RISE program.

### Factors associated with VL re-suppression

In the unadjusted model analysis, only age, enrollment setting, ART regimen line, and ART duration were significantly associated with VL re-suppression (Table 3). PLHIV in the 10–19 years age band were 13% more likely to be re-suppressed compared to the 0–9 years (CRR: 1.13; 95% CI 1.04 to 1.23). Similarly, PLHIV in the 20–29 years (CRR: 1.02; 95% CI 0.96 to 1.08), 30–39 years (CRR: 1.04; 95% CI 0.98 to 1.11) and > = 40 years (CRR: 1.11; 95% CI 0.96 to 1.07) age bands were more likely to be re-suppressed; however, these were not statistically significant. PLHIV enrolled in the facility were 5.6% more likely to be resuppressed post-EAC than those enrolled in community settings (CRR: 1.06; 95% CI 1.02 to 1.10). Conversely,

**Table 1. Demographic and clinical characteristics of People living with HIV (PLHIV) with unsuppressed VL (>1000 copies/ml of blood) who completed Enhanced Adherence Counselling (EAC).**

| Characteristics | N | (%) |
|---|---|---|
| **Total** | **1607** | |
| **Gender** | | |
| Female | 1054 | 65.59 |
| Male | 553 | 34.41 |
| **Age group** | | |
| 0–9 years | 114 | 7.09 |
| 10–19 years | 171 | 10.64 |
| 20–29 years | 226 | 14.06 |
| 30–39 years | 507 | 31.55 |
| > = 40 years | 589 | 36.65 |
| **Enrollment Settings** | | |
| Community | 197 | 12.26 |
| Facility | 1410 | 87.74 |
| **Clinical Staging** | | |
| Stage I | 1537 | 95.64 |
| Stage II | 62 | 3.86 |
| Stage III | 8 | 0.50 |
| **TB Co-infection** | | |
| No | 1557 | 96.89 |
| Yes | 50 | 3.11 |
| **ART Regimen Line** | | |
| First line | 1506 | 93.71 |
| Second Line | 101 | 6.29 |
| **Duration on ARV** | | |
| 0–2 years | 361 | 22.46 |
| 3–5 years | 550 | 34.23 |
| 6–10 years | 463 | 28.81 |
| >10 years | 233 | 14.50 |
| **Time to complete EAC sessions** | | |
| Median (IQR) | 12 (0–9 weeks) | |
| < = 12 weeks | 1516 | 94.34 |
| 13–24 weeks | 90 | 5.60 |
| ≥25 weeks | 1 | 0.06 |
| **Time from EAC session completion to repeat VL** | | |
| Median (IQR) | 8 (4–8 weeks) | |
| <2 weeks | 161 | 10.13 |
| 2–4 weeks | 140 | 8.81 |
| 4–8 weeks | 515 | 32.39 |
| 8–12 weeks | 243 | 15.28 |
| > = 12 weeks | 531 | 33.40 |

DTG-based regimens are the recommended first-line ARV drugs.

Protease inhibitor-based regimens are the recommended second-line ARV drugs.

**Table 2. Results showing the relationship between baseline characteristics and VL resuppression.**

| Characteristics | Detectable and suppressed <1000–50 copies/ml | VL undetectable <50 copies/ml | Total Suppressed <1000 copies/ml | Unsuppressed > = 1000 copies/ml | P value |
|---|---|---|---|---|---|
| | N (%) | N (%) | N (%) | N (%) | |
| **Total** | **336** | **1118** | **1454** | **153** | |
| **Gender** | | | | | |
| Female | 219(22.91) | 737(77.09) | 956 (90.70) | 98 (9.30) | 0.6742 |
| Male | 117(23.49) | 381(76.91) | 498 (9.05) | 55 (9.95) | |
| **Age group** | | | | | |
| 0–9 years | 39(36.79) | 67(63.21) | 106 (92.98) | 8 (7.02) | |
| 10–19 years | 32(22.70) | 109(77.30) | 141 (82.46) | 30 (17.54) | < .0031 |
| 20–29 years | 45(21.63) | 163(78.37) | 208 (92.04) | 18 (7.96) | |
| 30–39 years | 98(21.44) | 359(78.56) | 457 (90.14) | 50 (9.86) | |
| > = 40 years | 122(22.51) | 420(77.49) | 542 (92.02) | 47 (7.98) | |
| **Enrollment Settings** | | | | | |
| Community | 20(10.70) | 167(89.30) | 187 (94.92) | 10 (5.08) | |
| Facility | 316(24.94) | 951(75.06) | 1267 (89.86) | 143 (10.14) | < .0233 |
| **Clinical Staging** | | | | | |
| Stage I | 324(23.19) | 1073(76.81) | 1397 (90.89) | 140 (9.11) | |
| Stage II | 11(21.15) | 41(78.85) | 52 (83.87) | 10 (16.13) | 0.0047 |
| Stage III | 1(20.00) | 4(80.00) | 5 (62.50) | 3 (37.50) | |
| **TB Co-infection** | | | | | |
| No | 332(23.55) | 1078(76.45) | 1410 (90.56) | 147 (9.44) | < .0001 |
| Yes | 4(9.09) | 40(90.91) | 44 (88.00) | 6 (12.00) | |
| **ART Regimen Line** | 15(32.61) | 31(67.39) | | | |
| First line | 321(22.80) | 1087(77.20) | 1408 (93.49) | 55 (54.46) | < .0001 |
| Second Line | 15(32.61) | 31(67.39) | 46 (45.54) | 98 (6.51) | |
| **Duration on ARV** | | | | | |
| 0–2 years | 66(19.64) | 270(80.36) | 336 (93.07) | 25 (6.93) | |
| 3–5 years | 102(20.32) | 400(79.68) | 502 (91.27) | 48 (8.73) | < .0259 |
| 6–10 years | 115(27.64) | 301(72.36) | 416 (89.85) | 47 (10.15) | |
| >10 years | 53(26.50) | 147(73.50) | 200 (85.84) | 33 (14.16) | |

PLHIV on second-line ART regimens were 49% less likely to be re-suppressed when compared to PLHIV who were on first-line ART regimens at the time of analysis (CRR: 0.49; 95% CI 0.39 to 0.60). Our analysis found that the likelihood of re-suppression post-EAC increased with a longer time on treatment: 3–5 years (CRR: 1.02; 95% CI 0.98 to 1.60); 6–10 years (CRR: 1.04; 95% CI 0.99 to 1.08); and >10 years (CRR: 1.08; 95% CI 1.02 to 1.15) respectively. PLHIV who had been on ART treatment for more than 10 years had a statistically significant 8% higher likelihood of re-suppression than PLHIV who had been on ART treatment for no more than 2 years.

Our adjusted model found age group and ART regimen line as the two factors significantly associated with viral re-suppression (Table 3). PLHIV in the 10–19 years age band were 10% more likely to be re-suppressed compared to the 0–9 years (ARR: 1.10; 95% CI 1.01 to 1.19), which was slightly lower than the probability from our bivariate analysis. While all other age bands had a reduced likelihood of re-suppression, this was not statistically significant. Notably,

**Table 3. Demographic and clinical characteristics associated with resuppression of VL after completion of enhanced adherence counseling.**

| Characteristics | CRR (95% CI) | ARR (95% CI) | P value |
|---|---|---|---|
| **Total** | | | |
| **Gender** | | | |
| Female | reference | reference | |
| Male | 1.0072 (0.9737 -1.0418) | 1.0031 (0.9716–1.0357) | 0.8472 |
| **Age group** | | | |
| 0–9 years | reference | reference | |
| 10–19 years | 1.1305 (1.0378–1.2313) * | 1.0958 (1.0101–1.1886) | **0.0276** |
| 20–29 years | 1.0174 (0.9553–1.0836) | 0.9969 (0.9361–1.0617) | 0.9231 |
| 30–39 years | 1.0431 (0.9836–1.1063) | 0.9948 (0.941–1.0518) | 0.8556 |
| > = 40 years | 1.1061 (0.9608–1.0746) | 0.9774 (0.9264–1.0312) | 0.4024 |
| **Enrollment Settings** | | | |
| Community | reference | reference | |
| Facility | 1.0564 (1.0183 -1.0959) * | 1.0088 (0.9725–1.0465) | 0.6390 |
| **Clinical Staging** | | | |
| Stage I | reference | reference | |
| Stage II | 1.0837 (0.9705 -1.2101) | 1.0759 (0.9687–1.195) | 0.1717 |
| Stage III | 1.4543 (0.8500–2.4880) | 1.2497 (0.7671–2.0361) | 0.3707 |
| **TB Co-infection** | | | |
| No | reference | reference | |
| Yes | 1.0291 (0.9278–1.1414) | 1.0353 (0.9469–1.1319) | 0.4461 |
| **ART Regimen Line** | | | |
| First line | reference | reference | |
| Second Line | 0.4871 (0.3934 -0.6032) * | 0.5043 (0.4082–0.6229) | **< .0001** |
| **Duration on ARV** | | | |
| 0–2 years | reference | reference | |
| 3–5 years | 1.0197 (0.9815 -1.0595) | 0.9953 (0.959–1.033) | 0.8039 |
| 6–10 years | 1.0359 (0.9937–1.0799) | 0.9962 (0.9561–1.0381) | 0.8572 |
| >10 years | 1.0843 (1.0219 -1.1505) * | 1.0017 (0.9461–1.0605) | 0.9543 |

CRR = crude risk ratio; ARR = adjusted risk ratio

* = p < 0.05.

there was a 50% reduced likelihood of viral re-suppression among PLHIV on second-line regimens compared to PLHIV on first-line regimens (ARR: 0.50; 95% CI 0.41 to 0.62).

## Discussion

Our study explored the factors associated with VL re-suppression among 1,607 PLHIV who completed EAC in RISE-supported facilities across five states in Nigeria. After completing EAC, 91% of participants achieved a suppressed VL post-EAC, with 23% having VL between > = 50 and < 1000 copies/ml and 76% having an undetectable VL. Factors associated with VL re-suppression were age and ART regimen line. Our rate of re-suppression is similar to results from other studies that showed rates of re-suppression of 90%-95% after EAC [9,13,18]. Re-suppression rates depend on EAC's type, intensity, and documentation [13,19]. Compared to findings from other similar studies, our 91% re-suppression rate was the highest [20–24].

The demographic and clinical characteristics of the PLHIV in this study provided further insights into the factors associated with VL re-suppression. Age was found to be associated with VL re-suppression, with PLHIV aged 10–19 years significantly more likely to be re-suppressed post-EAC. This finding suggests that this age band may be associated with better post-EAC adherence and treatment outcomes. Our results may be explained by the immediate enrollment on Operation Triple Zero (OTZ), intensive follow-up of such children and adolescents, and caregiver involvement in care by the RISE team. Unlike our findings around age, other studies have found that adolescents were less likely to be re-suppressed post-EAC [9,24]. Similarly, previous studies have found that children and adolescents had comparatively lower suppression rates, with a higher probability of viral suppression among older adolescents [25,26]. Alongside poor adherence, this noted sub-optimal suppression among children and adolescents has been attributed to challenges around access to health facilities, satisfaction with healthcare services, and psychosocial factors, including stigma, school-related activities, caregiver status, knowledge, and availability [27–30].

The overall proportion of PLHIV with VL re-suppression did not differ between PLHIV enrolled in health facilities compared to those in the community. Community ART service delivery has been noted to improve ART uptake, retention, and viral suppression [31–33]. However, a study assessing facility and community refill strategies after community enrollment has noted no difference in viral suppression [34]. Moreover, the provision of comprehensive HIV care services in community settings where PLHIV may have relatively better access to counseling, support, and regular monitoring is effective in increasing HIV treatment coverage, and an optimal combination of both community and facility approaches will be vital in improving treatment outcomes [35].

Similar to the enrollment settings above, PLHIV who had been on ART for over 10 years were significantly more likely to be re-suppressed from our bivariate analysis but not in our multivariate analysis. VL suppression has been generally associated with increasing duration of treatment and adherence [26,36–38]. However, a study conducted in Ethiopia on VL among PLHIV post-EAC noted that PLHIV on ART treatment between 3–5 years were less likely to be re-suppressed, unlike ours [18].

ART regimen was a significant predictor in our bivariate and multivariate analyses. PLHIV on second-line ART regimens were less likely to be resuppressed. This could be due to the majority 1,376 (91%) of PLHIV being on DTG-based first-line regimens per the Nigeria National Treatment Guidelines [39]. DTG-based regimens have been proven to be highly productive in achieving virological suppression in PLHIV [38,40,41]. In addition, the prior viral unsuppressed results may have been due to non-adherence, and with improved adherence following the EAC sessions, we would expect comparatively better viral re-suppression.

Furthermore, with the introduction and transition of PLHIV to DTG-based regimens, PLHIV who had been on second-line regimens may have never benefitted from the new DTG-based regimens or may have been transitioned back to the new DTG-based regimens. However, two studies on virological outcomes post-EAC in Nigeria and Zimbabwe have noted a higher likelihood of viral re-suppression among PLHIV on second-line ART regimens [2,21]. This has been attributed to possible lower resistance to second-line drugs and PLHIV's "last chance" perception of medically responsive therapy [21]. Moreover, unsuppressed VL post-EAC has been associated with prolonged first-line ART regimen treatment and drug resistance [13].

The strength of this study is the use of routine program data, which provides valuable insights into the real-world effectiveness of EAC in Nigeria. However, there are several limitations to consider. First, the data used was cross-sectional, so causality cannot be inferred from the findings. Second, the analysis relied on routine program data, which may be subject to

missing or incomplete information. Additionally, the analysis was limited to only those variables that are routinely collected from patients during visits and available in the EMR. We did not measure other potential vital factors that may be associated with VL re-suppression, such as socioeconomic factors, social support, or mental health status.

In conclusion, this study highlights the effectiveness of EAC in promoting VL suppression in routine program settings, with 91% re-suppression. VL suppression after EAC completion on the RISE project surpasses the WHO 70% target. The findings suggest that EAC can improve adherence to antiretroviral therapy and achieve the UNAIDS global target of 95% VL suppression. The study also identified essential factors associated with VL re-suppression, including age and ART regimen type. These findings have implications for programmatic and clinical decision-making and highlight the need for tailored interventions.

## Supporting information

**S1 Data.**
(XLSX)

## Acknowledgments

Special thanks are due to Dr. Omari Habib, for his guidance in analyzing the adjusted model for the Log binomial regression using SAS 9.4 statistical software. His contributions were instrumental in the successful completion of this research.

## Author Contributions

**Conceptualization:** Gbenga Benjamin Obasa.

**Data curation:** Gbenga Benjamin Obasa, Mukhtar Ijaiya, Ejike Okwor.

**Formal analysis:** Gbenga Benjamin Obasa, Remi Oladigbolu.

**Investigation:** Kelly Curran.

**Methodology:** Gbenga Benjamin Obasa.

**Project administration:** Babafemi Dare, Franklin Emerenini, Prince Anyanwu, Yemisi Ogundare, Emmanuel Atuma.

**Resources:** Gbenga Benjamin Obasa, Mukhtar Ijaiya.

**Software:** Gbenga Benjamin Obasa.

**Supervision:** Adewale Akinjeji, Molly Strachan, Ruby Fayorsey, Kelly Curran.

**Validation:** Ruby Fayorsey.

**Writing – original draft:** Gbenga Benjamin Obasa.

**Writing – review & editing:** Gbenga Benjamin Obasa, Mukhtar Ijaiya, Kate Brickson, Jennifer Zech, Molly Strachan, Ruby Fayorsey.

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
