## [Decision Letter · Decision Letter 0]

18 Mar 2024

PGPH-D-24-00075

Factors associated with viral Load re-suppression after enhanced adherence counseling among recipients of care with an Initial high viral load result in RISE-supported facilities in five States in Nigeria

Dear Dr. Obasa,

Thank you for submitting your manuscript to PLOS Global Public Health. After careful consideration, we feel that it has merit but does not fully meet PLOS Global Public Health’s publication criteria as it currently stands. Therefore, we invite you to submit a revised version of the manuscript that addresses the points raised during the review process.

We look forward to receiving your revised manuscript.

Kind regards,

Humayun Kabir

Academic Editor

Journal Requirements:

1. Please provide additional details regarding participant consent. In the ethics statement in the Methods and online submission information, please ensure that you have specified (1) whether consent was informed and (2) what type you obtained (for instance, written or verbal, and if verbal, how it was documented and witnessed). If your study included minors, state whether you obtained consent from parents or guardians. If the need for consent was waived by the ethics committee, please include this information.

3. Please amend your detailed Financial Disclosure statement. This is published with the article. It must therefore be completed in full sentences and contain the exact wording you wish to be published.

If you did not receive any funding for this study, please simply state: “The authors received no specific funding for this work.

4. In the online submission form, you indicated that "The datasets used and analyzed during the current study are available from the corresponding author upon reasonable request. This analysis was carried out using data from Reaching Impact, Saturation, and Epidemic (RISE) Control program funded by the U.S. President's Emergency Plan for AIDS Relief (PEPFAR) through the United States Agency for International Development (USAID)". All PLOS journals now require all data underlying the findings described in their manuscript to be freely available to other researchers, either 1. In a public repository, 2. Within the manuscript itself, or 3. Uploaded as supplementary information.

Additional Editor Comments (if provided):

Reviewers' comments:

Reviewer's Responses to Questions

**Comments to the Author**

1. Does this manuscript meet PLOS Global Public Health’s publication criteria? Is the manuscript technically sound, and do the data support the conclusions? The manuscript must describe methodologically and ethically rigorous research with conclusions that are appropriately drawn based on the data presented.

Reviewer #1: No

Reviewer #2: Yes

2. Has the statistical analysis been performed appropriately and rigorously?

Reviewer #1: No

Reviewer #2: Yes

3. Have the authors made all data underlying the findings in their manuscript fully available (please refer to the Data Availability Statement at the start of the manuscript PDF file)?

Reviewer #1: No

Reviewer #2: Yes

4. Is the manuscript presented in an intelligible fashion and written in standard English?

Reviewer #1: No

Reviewer #2: Yes

5. Review Comments to the Author

Reviewer #1: Title

Remove acronym from the title

Abstract

- Re-write the abstract. It is not well structured. Background information is mixed up with methods in the current version of the abstract.

- Write ROC in full.

- Results showed a 91% re-suppression …. Please use absolute numbers before proportions.

- Adolescents (10-19 years) had a higher likelihood of re-suppression, while ROC on second-line regimens exhibited lower resuppression rates…. In this sentence, if you are reporting 2 separate outcomes, please break up the sentence. It is confusing to the reader in the current form.

Study design

A descriptive study using routinely collected service delivery…. What type of descriptive study? Was it cross-sectional survey, retrospective cohort or. Please specify….

Program data and setting

These sections are not clear. Not written well and I can not understand the data.

Study population

The information in this section is the same as that in study setting. Consider revising.

Data analysis

Not well written. For example, sentence In addition, Pearson's Chi-Square was used to compare demographics, clinical 147 characteristics, and VL re-suppression with a p-value of 0.05.

Results

Not well presented. Not speaking to the objectives. Study population not well described and shown in the results.

General comments

The paper will benefit from editing. There is need to improve cohesion and logical flow of the paper.

ROC is not standard acronym. Please remove

There is so much use of acronyms. Most not standard.

Does not follow STROBE guidelines.

Reviewer #2: Methods:

Give context of how is usual HIV care provided in Nigeria (facilities, ART programs).

Description of RISE-supported Health Facilities (are these new facilities developed for the RISe program or integrated in existing infrastucture)

Describe what exactly was the EAC (intervention) that was provided. Who is providing the EAC (layperson, medic)?Was it the same for children, adolescents, women and elderly?

Results:

What was the baseline Viral load suppression ?

Were there any patients who received no EAC (Control group)?

The patients included in the study had they completed all sessions for EAC, what about those who completed done or two sessions for EAC?

Discussion:

The study results are higher (90% vs 54%) and opposite (higher in adolescents) than other studies. Comment on what could be the reasons for these differences.

The authors need to be cautious in claiming effectiveness without baseline or controls.

6. PLOS authors have the option to publish the peer review history of their article (what does this mean?). If published, this will include your full peer review and any attached files.

**Do you want your identity to be public for this peer review?** For information about this choice, including consent withdrawal, please see our Privacy Policy.

Reviewer #1: No

Reviewer #2: **Yes: **Natasha Shaukat

---

## [Decision Letter · Decision Letter 1]

5 Sep 2024

Factors associated with viral Load re-suppression after enhanced adherence counseling among People Living with HIV with an Initial high viral load result in selected Nigerian States.

PGPH-D-24-00075R1

Dear Mr. Obasa,

We are pleased to inform you that your manuscript 'Factors associated with viral Load re-suppression after enhanced adherence counseling among People Living with HIV with an Initial high viral load result in selected Nigerian States.' has been provisionally accepted for publication in PLOS Global Public Health.

Best regards,

Julia Robinson

Executive Editor

Reviewer Comments (if any, and for reference):

Reviewer's Responses to Questions

**Comments to the Author**

1. If the authors have adequately addressed your comments raised in a previous round of review and you feel that this manuscript is now acceptable for publication, you may indicate that here to bypass the “Comments to the Author” section, enter your conflict of interest statement in the “Confidential to Editor” section, and submit your "Accept" recommendation.

Reviewer #2: All comments have been addressed

2. Does this manuscript meet PLOS Global Public Health’s publication criteria? Is the manuscript technically sound, and do the data support the conclusions? The manuscript must describe methodologically and ethically rigorous research with conclusions that are appropriately drawn based on the data presented.

Reviewer #2: Yes

3. Has the statistical analysis been performed appropriately and rigorously?

Reviewer #2: Yes

4. Have the authors made all data underlying the findings in their manuscript fully available (please refer to the Data Availability Statement at the start of the manuscript PDF file)?

Reviewer #2: Yes

5. Is the manuscript presented in an intelligible fashion and written in standard English?

Reviewer #2: Yes

6. Review Comments to the Author

Reviewer #2: NA

7. PLOS authors have the option to publish the peer review history of their article (what does this mean?). If published, this will include your full peer review and any attached files.

**Do you want your identity to be public for this peer review?** For information about this choice, including consent withdrawal, please see our Privacy Policy.

Reviewer #2: No
